# Circulating Extracellular Vesicles As Biomarkers and Drug Delivery Vehicles in Cardiovascular Diseases

**DOI:** 10.3390/biom11030388

**Published:** 2021-03-05

**Authors:** Renata Caroline Costa de Freitas, Rosario Dominguez Crespo Hirata, Mario Hiroyuki Hirata, Elena Aikawa

**Affiliations:** 1Center for Interdisciplinary Cardiovascular Sciences, Division of Cardiovascular Medicine, Brigham and Women’s Hospital, Harvard Medical School, Boston, MA 02115, USA; rcfreitas@bwh.harvard.edu; 2Department of Clinical and Toxicological Analyses, School of Pharmaceutical Sciences, University of Sao Paulo, Sao Paulo 05508-000, Brazil; rosariohirata@usp.br (R.D.C.H.); mhhirata@usp.br (M.H.H.); 3Center for Excellence in Vascular Biology, Division of Cardiovascular Medicine, Brigham and Women’s Hospital, Harvard Medical School, Boston, MA 02115, USA; 4Department of Human Pathology, Sechenov First Moscow State Medical University, 119992 Moscow, Russia

**Keywords:** extracellular vesicles, exosomes, ectosomes, biomarkers, RNA, proteins, lipids, cardiovascular disease

## Abstract

Extracellular vesicles (EVs) are composed of a lipid bilayer containing transmembrane and soluble proteins. Subtypes of EVs include ectosomes (microparticles/microvesicles), exosomes, and apoptotic bodies that can be released by various tissues into biological fluids. EV cargo can modulate physiological and pathological processes in recipient cells through near- and long-distance intercellular communication. Recent studies have shown that origin, amount, and internal cargos (nucleic acids, proteins, and lipids) of EVs are variable under different pathological conditions, including cardiovascular diseases (CVD). The early detection and management of CVD reduce premature morbidity and mortality. Circulating EVs have attracted great interest as a potential biomarker for diagnostics and follow-up of CVD. This review highlights the role of circulating EVs as biomarkers for diagnosis, prognosis, and therapeutic follow-up of CVD, and also for drug delivery. Despite the great potential of EVs as a tool to study the pathophysiology of CVD, further studies are needed to increase the spectrum of EV-associated applications.

## 1. Introduction

Extracellular vesicles (EVs) is a generic term for particles naturally released from the cells that are delimited by a lipid bilayer-containing transmembrane and soluble proteins and cannot replicate, according to The International Society for Extracellular Vesicles (ISEV) [1]. The first study that reported EVs was published in 1967 describing EVs as minute dust-like particulate material rich in lipid content [2]. 

EVs can be classified based on size as small EVs (sEVs), with range < 100 nm or < 200 nm, and medium/large EVs (m/lEVs), with size range > 200 nm [1]. EVs also can be classified based on cell origin as ectosomes (microparticles/microvesicles), exosomes, and apoptotic bodies. Ectosomes (size range 100–500 nm) are released from the plasma membrane budding, exosomes (size range 50–150 nm) are assembled from the endosomal pathway and released by exocytosis of multivesicular bodies (MVB), and apoptotic bodies (size range 500 nm–2 µm) are generated during apoptotic cell shrinkage and death [3,4,5,6]. There are various methods used for isolation of EVs or a specific EV subtype that have been recently reviewed [7], such as ultracentrifugation (UC), size-exclusion chromatography (SEC), filtration, immunoaffinity-based isolation, commercial reagents (using polymers), microfluidics, and asymmetric flow field-flow fractionation (AF4). To increase the specificity or purity, the methods can be combined.

EVs can be characterized by their cargos and surface protein biomarkers, including annexins (e.g., annexin 1, 5, 6, and 11), disintegrin and metalloproteinase domain-containing protein 10 (ADAM10), angiotensin-converting enzyme (ACE), EH domain-containing protein 4 (EHD4), major histocompatibility complex class II (MHC II), flotillin-1 (FLOT1), and heat-shock 70-kDA (HSC70/HSP73, HSP70/HSP72). Other proteins are used as exosome markers, such as tetraspanins (CD9, CD63, CD81, and CD82), stress proteins (Hsc70 and Hsp90), proteins involved in membrane fusion (Rabs, and ARF6), and protein members of the endosomal sorting complex required for transport (Alix and TSG101) [8,9]. Microvesicles have content similar to exosomes that include specific proteins, such as integrins, glycoproteins, and metalloproteinases [8,10]. To identify EV’s protein markers, the main methods include Western blotting, ELISA, flow cytometry (FCM), and nano-FCM. In addition, transmission electron microscopy (TEM), dynamic light scattering (DLS), and nanoparticle tracking analysis (NTA) are commonly used [7,11,12].

EVs have emerged as possible biomarker sources from several diseases, due their ability to modulate near- or in long-distance intercellular communication influencing the disease development and progression [13,14,15]. Intercellular communication consists of transferring EV bioactive cargos or activating signaling pathways to recipient cells, which can lead to phenotypic and functional changes in their target cells [5,16]. EVs are present in various tissues and biological fluids from which they can be recovered and monitored in both physiological and pathological conditions [17,18]. The quantity, origin, and internal cargo (e.g., nucleic acids, proteins, and lipids from parental cells) are variable in different pathophysiological processes [14,19]. EVs also have a metabolically active outer membrane that protects their content until released into recipient cells [17]. 

Circulating EVs have attracted great interest in the field of cardiovascular medicine due to their high stability. EVs offer a non-invasive access to monitor the status of the cardiovascular diseases (CVD), and the use of circulating EVs as diagnostic biomarkers [13,20]. CVD causes the highest number of deaths and vast health and economic burdens worldwide [21,22]. CVD include several pathologies such as coronary artery disease (CAD), cerebrovascular disease, peripheral arterial disease, ischemic heart disease, hypertension, and heart failure (HF). Early detection and management of CVD can decrease the risk of heart attack and stroke in individuals at high risk of CVD, and, therefore, reduce premature morbidity and mortality [23].

There has been a growing interest in exploring the EVs in the diagnostic, prognostic, and therapeutic monitoring of CVD, as well as drug delivery (Figure 1). This review discusses the role of circulating EVs in CVD based on origin, amount, and content of the EVs, and highlights their application as biomarkers and drug delivery tool in several cardiovascular pathologies.

## 2. Origins of Extracellular Vesicles Related to CVD

Circulating EVs are released by almost all cells, including cardiovascular system-related cells (e.g., blood, heart, and blood vessels) [24,25]. Biologic fluids, such as blood [26], urine [27], saliva [28], breast milk [29], and seminal fluid [30], as well as conditioned media from cell culture experiments [31,32], all contain EVs.

### 2.1. Blood-Cells Derived EVs

EVs can be released from platelets (and megakaryocytes), erythrocytes, and leukocytes. The main sources of circulating EVs are platelets, which are derived from megakaryocytes, and are regulators of hemostasis, inflammation, and vascular integrity [33,34]. Some reviews [35,36,37] have reported the role of platelet-derived EVs in atherosclerosis, acute coronary syndrome (ACS), and thrombosis, being considered as potential EV source in CVD [38]. Platelet EVs have procoagulant and pro-inflammatory effects [39,40,41,42], and serve as important messengers, communicating the changes that occur in the plasma to bone marrow cells [43] and other tissues impermeable to platelets [33]. 

Circulating EVs derived from erythrocytes are released to clear away harmful molecules and prevent the early removal of these cells from circulation [44,45]. Erythrocyte-derived EVs are also shown to be associated with CVD. Patients with ST-segment elevation myocardial infarction (STEMI) who undergo angioplasty have approximately double of erythrocyte-derived EVs as compared to healthy subjects [46]. These EVs are also associated with atherosclerosis by inducing hypercoagulation, inflammation and cell adhesion [47,48].

Leukocyte-derived EVs can originate from neutrophils, monocytes/macrophages, and lymphocytes, as differentiated by specific markers associated with their parental cells [49]. EVs released by leukocytes may have an important role in maintaining or disrupting vascular homeostasis and pathological thrombosis contributing to inflammatory responses [49]. T cell-derived EVs were increased in the circulation of an animal model of angiotensin II (ANG II)-induced hypertension, resulting in inflammatory response [50]. Plasma levels of the leukocyte-derived EVs were elevated in patients with hypertension and hyperlipidemia [51]. Melnikov et al. [52] identified monocytes-derived EVs carrying monomeric C-reactive protein (mCRP) in the blood that was associated with inflammatory status in CAD patients. 

### 2.2. Heart Cell- and Blood Vessel-Derived EVs

It has been reported that EVs could be released from major cell types in the heart [15,53], such as cardiomyocytes, fibroblasts, endothelial cells, and vascular smooth muscle cells (SMC). Loyer et al. [54] demonstrated, using a murine model of myocardial infarction, that EVs released by cardiomyocytes and endothelial cells following myocardial infarction could be taken up by monocytes and regulate the cardiac inflammatory response by releasing of proinflammatory cytokines.

Endothelial EVs are associated with the progression of atherosclerosis [55], hypertension [51], and CAD [56]. On the other hand, EVs can play protective roles. For example, endothelial cell-derived EVs were reported as cardioprotective molecules releasing proteins involved in cellular homeostasis and preservation in the ischemia-reperfusion injury in a chip model of human heart [57]. Several studies were summarized in a review that reported activated endothelial cell-derived EVs were also involved in the regulation of cardiac and vascular remodeling in HF [58]. 

Cardiomyocyte-derived EVs take an important part in the progression of CVD, because they can carry a wide variety of biomolecules, such as proteins and miRNAs, to other cell types and regulate the function and gene expression in these cells [59], especially promoting cardiac repair [60]. Cardiomyocyte-derived EVs secreted from primary cardiomyocytes and human induced pluripotent stem cell-derived cardiomyocytes (hiPSC-CM) can have angiogenic effects after myocardial infarction through inducing increase expression of miRNAs and proteins, such as growth factors [61,62], also inducing cardiac fibrosis by release of specific miRNAs via myocyte-fibroblast cross-talk [63]. EVs released from cardiomyocytes derived from human-induced pluripotent stem cells were also used in treatment of heart injury, including myocardial infarction, contributing to cardiac regeneration, through cardiac-specific miRNAs activity [64,65].

Cardiac fibroblast-derived EVs stimulated cell cardiac migration [66], SMC proliferation, and vascular remodeling [67] by release of miRNAs. Vascular SMC-derived EVs are enriched with RNAs, proteins and lipids associated with vascular remodeling, calcification and coagulation [68,69,70,71,72], familial hypercholesterolemia, and CAD [73]. Most studies have evaluated changes in SMC caused by EVs derived from other cells. For example, EVs derived from bone marrow mesenchymal stem cells (BMSC) have been shown to induce calcification in vascular SMC by modifying miRNA profiles [74], and EVs derived from platelets could modulate inflammatory response in vascular SMC by presenting chemokine CXCL4 and membrane-bound effectors [75]. 

### 2.3. EVs Interaction between Cells from Different Origins

The role of EVs in intercellular communication and interaction between heart-derived cells was reviewed by Hafiane et al. [8], and EV communication between platelets, monocytes, and endothelial cells was associated with myocardial ischemia. Weiss et al. [76] reported differential interaction of platelet-derived EVs with monocytes and other leukocytes, which were identified by specific markers using flow cytometry. The authors used CD41 as marker of platelet origin, CD45-PB and CD14-PE as monocyte markers, CD16/56-PC5 as granulocyte and NK cell marker, and CD3-ECD as T cell marker. 

Quiescent endothelial cells were shown to release EVs that were able to suppress monocyte activation and anti-inflammatory molecules associated with vascular inflammation in CVD [77]. TNF-α-induced inflamed endothelial cells were shown to release EVs enriched in cytokines, chemokines and other inflammatory markers, which when transferred to monocytes promoted their differentiation to pro- or anti-inflammatory phenotypes [78]. 

EVs derived from macrophage foam cells from patients with atherosclerosis were shown to integrate into vascular SMC and induce their migration and adhesion [79]. EVs also participate in communication between endothelial and vascular SMC. Boyer et al. [80] demonstrated that endothelial-derived EVs could also stimulate protein synthesis and senescence of vascular SMC. In addition, a recent study reported EV-mediated transmission of RNA between endothelial cells and SMC, alleviating ANG II-induced vascular dysfunction [81]. 

## 3. Extracellular Vesicles Quantification as Biomarker in CVD

Several studies have shown an association of circulating EV counts with CVD, suggesting a potential application of EV quantification as a biomarker for diagnostic and therapeutic monitoring [25,82,83]. Although using EV counts from particular cell type as biomarker seems promising, the major limitation of this approach is the lack of standardization of methods, resulting in difficulty to compare studies from multiple research groups [25].

The release of platelet-derived EVs was shown to be increased in plasma, under conditions with enhanced platelet activation, such as myocardial infarction and exposure to modified lipoproteins [33,84]. Likewise, in arterial and venous thrombosis, the activated platelets increase the circulating EV counts compared with healthy condition [25]. 

Patients with atherothrombotic diseases and atherosclerotic lesions have high levels of circulating EVs derived from endothelial cells, vascular SMC, platelets, leukocytes or erythrocytes [85]. Sansone et al. observed an increase of endothelial-derived EVs in the plasma of patients with arterial hypertension with and without CAD [56]. Plasma levels of leukocyte-derived EVs were reported to be increased in atherosclerotic patients, and they were correlated with the progression of the atherosclerosis [79].

Several studies have reported increased counts of EVs in ACS conditions [86,87,88,89,90]. Serum EVs were found to be higher in patients with STEMI than whose with stable angina or control subjects, suggesting early stages increases in the disease due to thrombus formation and ischemia-induced stress [91]. Erythrocyte-derived EV counts were also elevated in STEMI patients [46]. 

Importantly, the increase of circulating EVs can be detected shortly after the pathological stimulus. Deddens et al. [92] demonstrated that plasma EVs are rapidly detectable. In one study, the amount of EVs was already increased one hour after myocardial infarction. Ge et al. [93] observed a significant increase in heart tissue EVs release 24 h after myocardial ischemia/reperfusion (I/R). 

Patients with persistent atrial fibrillation (AF) and a high level of inflammation showed markedly increased EV concentration compared to subjects without AF [82]. In addition, the inflammation contributes to platelet activation that induces the release of EVs in a prothrombotic state [82]. A recent study also showed that circulating EVs were increased in patients with AF and a higher risk of stroke than non-AF patients of similar age [94].

Circulating EVs derived from endothelial cells were explored in a prospective study, which demonstrated that patients with HF had increased plasma levels of endothelium-derived microparticles compared to healthy subjects [95]. These HF patients had a higher probability of cardiovascular events (e.g., cardiovascular death, non-fatal myocardial infarction, ischemic stroke, or re-hospitalization related to HF), and it was suggested that EV counts could be a useful prognostic biomarker. Patients with symptoms of chronic HF had increased number of circulating endothelial-derived EVs that were correlated with increase of mortality and recurrent hospitalization risk due to HF [96]. HF patients also had increased serum levels of EVs compared to healthy subjects [97]. A recent review [58] has reported that the number of EVs might be important to differentiate the severity of HF.

Circulating EV counts are also altered in patients with metabolic disorders that increase the risk of CVD. For example, the total number of circulating EVs was shown to be higher in patients with metabolic syndrome (MetS) compared to non-MetS subjects [98]. Increased levels of endothelial-derived EVs were also observed in diabetic patients compared with healthy controls, and they were closely associated with vascular dysfunction [99]. Circulating levels of lymphocyte-derived EVs were also increased in patients with familial hypercholesterolemia [100]. 

## 4. Extracellular Vesicle as Biomarkers in CVD

Studies on the important regulatory effects of EVs in CVD has been motivated due to EV stability, their specific signatures associated with cell activation or injury, and their intrinsic activity and immunomodulatory properties [13]. The changes in EV cargo, including RNAs, proteins, and lipids, as potential biomarkers in CVD are reviewed in Table 1.

### 4.1. Extracellular Vesicles Carrying RNAs

The EV transcriptome of various cell types is important due to the biological relevance of RNA activity in several cardiovascular pathologies [58,109,125,126]. EVs are carriers of various RNA types, such as messenger RNA (mRNA), transfer RNA (tRNA), small interference RNA (siRNA), long-non-coding RNA (lncRNA), and microRNA (miRNA) [13]. An earlier study identified mRNAs and miRNAs in EVs by microarray technology and showed the transference of functional RNA between three cell lines [127].

Kenneweg et al. [101] showed lncRNA-enriched EVs in cardiac ischemia. In this context, lncRNA *Neat1* was necessary for fibroblast and cardiomyocyte survival, and the silencing of *Neat1* resulted in reduced heart function after myocardial infarction. A study identified 185 differentially expressed circular RNAs (circRNAs), covalently closed RNAs, involved in the metabolic process from EVs of the murine heart post-I/R injury compared with control, and these circRNAs may regulate target genes by acting on the miRNAs [93].

miRNAs are short non-coding RNAs (19-22 nucleotides) that regulate gene expression at the post-transcriptional level by binding to specific mRNAs with varying degrees of complementarity and leading to mRNA degradation and/or translational inhibition [128,129]. miRNAs control different physiological processes and abnormal patterns of expression have already been associated with many diseases [129]. Different types of cells can release miRNAs into the extracellular space in response to various stimuli and pathologies [130,131]. In peripheral circulation, EVs are responsible for protecting miRNAs from degradation by circulating ribonucleases [130,132]. 

The EV-miRNAs can be promising predictors or indicators for premature CVD detection. Increased expression of miR-126 and miR-199a isolated from circulating EVs was proposed to reduce the risk of major cardiovascular outcomes in patients with CAD [102]. Cheng et al. [133] suggested that expression of miR-126 and miR-21 could be used for early detection of CVD, such as acute myocardial infarction and FH. Another study reported reduced plasma levels of EV-miR-126 in high-risk CVD patients and EV-miR-126 levels were negatively correlated with cardiac troponin I (cTnI) and N-terminal propeptide of B-type natriuretic peptide (NT-proBNP), suggesting miR-126 as a potential biomarker for CVD [103]. 

miRNA-208a expression was upregulated in the serum exosomes of ACS patients, and the study suggested its important role for the early diagnosis and prognosis of ACS [105]. Two EV-miRNA (miR-30e and miR-92a) that target ATP binding cassette (ABC)A1 were shown to be upregulated in plasma EVs from patients with coronary atherosclerosis [104]. Endothelial cells-derived EVs containing miR-92a were increased in patients with CAD, and this miRNA was shown to regulate angiogenesis in recipient endothelial cells [109]. EV-enriched miR-92a can be transferred from endothelial cells to macrophages and suppress Kruppel-like factor 4 (KLF4) expression in recipient cells, resulting in an atherosclerotic phenotype [110,134]. In addition, upregulation of the miR-1 in hepatocyte-derived EVs was associated with promotion of endothelial inflammation and facilitate atherogenesis by downregulation of KLF4 and activation of the NF-κB [135].

Increased levels of urinary EVs miRNAs were reported in patients with unstable CAD compared to whose with stable CAD [108]. The authors suggested an important role of miR-155 in disease progression that could be used as prognostic indicator and therapeutic target.

Atherogenic EVs from mouse and human macrophages were enriched in miR-146a, miR-128, miR-185, miR-365, and miR-503. Further, miR-146a was related to progression of atherosclerosis by decreasing cell migration and promoting macrophage entrapment in the vessel wall [111].

Elevated expression of miR-192, miR-194 and miR-34a in serum EVs was observed in HF patients after acute myocardial infarction [106]. Serum exosomal miR-92b-5p was increased in patients with HF due to dilated cardiomyopathy ant this miRNA was suggested as biomarker for diagnosis of HF [107]. 

EV miRNAs were also related to cardiovascular risk factors (i.e., diabetes, dyslipidemia, obesity, MetS). Cardiomyocytes isolated from type 2 diabetic rats had inhibitory effects on myocardial angiogenesis through the EV transfer of miR-320 into endothelial cells [136]. miR-24 and miR-130a were downregulated in plasma EVs of patients with familial hypercholesterolemia (FH), and miR-130a levels were inversely related to coronary atherosclerosis in suspected CAD patients, suggesting their role as potential biomarkers of FH and CAD [73].

A recent study using abdominal adipose tissue-derived mesenchymal stem/stromal cells showed four downregulated miRNAs (miR-136, miR-4798, miR-12,136, miR-222) and nine upregulated miRNAs (miR-630, miR-144, miR-143, miR-4787, miR-769, miR-8074, and miR-181a) from EVs of MetS patients. These deregulated miRNAs might control genes, which were associated with cellular senescence, cell cycle, metabolic processes, and apoptosis pathways [137].

### 4.2. Extracellular Vesicles Carrying Proteins 

Differences in EV protein levels occur in response to a variety of physiological or pathological stimuli. The protein profile might change already in a very early stage of the disease, which makes this content a potential early biomarker [115]. The EV protein cargo is heterogenous and dependent on the cell type or biofluid of origin [138].

EV proteins were suggested to be prognostic biomarkers of cardiovascular events. In this context, a prospective study demonstrated the increase of circulating CD31/Annexin 5-positive EVs as an independent predictor of cardiovascular risk in patients with stable CAD. High levels of CD31/Annexin 5 EVs were associated with higher incidence of death caused by CVD and higher need for revascularization [112,139].

A study of EV proteome of patients with myocardial infarction identified six novel EV protein markers of myocardial damage related to three pathways: complement activation (C1Q1A and C5), platelet activation (GP1BA and PPBP), and lipid metabolism (APOD and APOC3) [113]. Increased plasma levels of CD144-EVs were also suggested to be predictive of cardiovascular complications (i.e., ACS, ischemic stroke, revascularization, and death) in patients with high risk for CAD [114,139]. 

The link between EV proteins and atherosclerosis was described in a study, which showed that hypercholesterolemic patients with subclinical lipid-rich atherosclerotic plaques have a higher abundance of CD45/CD3-derived EVs than those in patients with fibrous plaques [100]. 

EV protein levels showed an association with stress-induced ischemia, especially proteins known to be related to inflammatory cascades such as SerpinC1, SerpinG1, CD14, and Cystatin C [115]. Serum EV proteins, such as cystatin C, polygenic immunoglobulin receptor (pIgR) and complement factor C5a (C5a), were suggested to be associated with ACS [117,140]. mCRP carried by monocyte-derived EVs was associated with inflammatory process in patients with CAD [52]. EVs can transport and delivery pro-inflammatory mCRP in endothelial cells [118]. mCRP carried by endothelial cell-derived EVs was also increased in patients with peripheral artery disease, and it was suggested to be a pro-inflammatory molecule and a potential indicator of vascular disease risk [119].

Plasma levels of EVs enriched in cystatin C, CD14, serpinG1, and serpinF2 were markedly increased in HF patients. These EVs proteins, previously related to systemic vascular events, were associated with high risk of HF in patients suspected of acute HF [116].

A recent study, using human cardiovascular cells, demonstrated that Annexin A1 induces EVs aggregation and microcalcification formation that promote CVD. These findings could lead to development of therapeutic strategies in CVD [120].

Urinary levels of EV proteins were decreased in patients with unstable CAD, however levels of CD45+ and CD11b+ EVs were increased and CD16+ EVs were decreased. These urinary EV proteins were suggested to be associated with CAD progression [108]. High levels of urinary EVs enriched in nephrin and podocalyxin were observed in patients with hypertension and these EV proteins were proposed to be useful diagnostic biomarkers [121]. Urinary EVs released by senescent nephron cells had increased levels of p16 (senescence marker) in patients with hypertension as compared to healthy volunteers. Urinary p16-positive EVs could serve as an early marker of nephron senescence and could be useful in disease management and therapeutic follow-up [122].

### 4.3. Extracellular Vesicles Carrying Lipids and Metabolites

Lipids are important components of vesicle bilayer membranes and specifics lipids, such as cholesterol and sphingomyelin, are enriched in vesicles compared to their parental cells and it might modulate recipient cell homeostasis [18,141]. Then, lipids are emerging as very important players for the physiological functions of these vesicles [142]. The first studies relating to the lipid composition of EVs were performed on prostate-derived EVs found in seminal fluid about twenty years ago [142,143]. The data have been included in the EVs databases such as Vesiclepedia [144], EVpedia [145], and Exocarta [146].

EVs lipids interact with receptors on the target cell and are thereafter internalized intro endosomes where they concentrate the bioactive lipids that they carry modulating endogenous cell lipid metabolism [18]. Since lipids are essential structural and functional constituents of EVs [142], the use of EV lipids as biomarkers of CVD may be promising, however, there are only a few studies on this topic. 

EVs can carry ceramides, sphingomyelin, lysophosphatidylcholine, arachidonic acid, and other fatty acids, cholesterol, prostaglandins, leukotrienes, and active lipolytic enzymes (such as phospholipase A2) on their membrane or within their lumen, and their lipid composition can be modified by in vitro manipulation [18]. Circulating EVs were enriched with different sphingolipid species (ceramides, dihydroceramides, and sphingomyelins) in patients with STEMI, and lipid levels correlated with cardiac troponin, leucocyte count, and lower left ventricular ejection fraction [123].

The amount of lipids in the shed EVs could be directly related to atherosclerosis, once accumulation of these lipids was associated with foam cell formation and apoptosis in macrophages mediated by toll-like receptors, which can lead to atherosclerosis [8,147]. EVs can be released by activated platelets, which are rich in phosphatidylserine, contributing in thrombin generation and promoting thrombosis [38,148]. Activated platelets also release EVs rich in arachidonic acid, which contributes to thrombosis in the recipient cells by the promotion of the cell adhesion and stimulation of prostacyclin and thromboxane A2 synthesis [149,150].

A pioneering study showed that urinary EV metabolites (4-aminohippuric acid, citric acid, and N-1-methylnicotinamide) were altered in patients with high cardiovascular risk. Urinary EV levels of 4-aminohippuric acid were increased, whereas citric acid and N-1-methylnicotinamide were reduced in patients with high cardiovascular risk, suggesting an important role of EV metabolites as biomarkers of CVD [124].

## 5. Extracellular Vesicles as Biomarkers for Therapeutic Responses in CVD

Plasma EV counts have been explored as biomarkers to assess the response to cholesterol-lowering and antiplatelet therapies. Suades et al. [151] showed a reduction in the number of circulating EVs, specifically microparticles, derived from endothelium, platelets, and inflammatory cells after lipid-lowering therapy with statins. Kulshreshtha et al. [83] also described that simvastatin reduced the secretion of EVs from various cell types. Conversely, atorvastatin was shown to increase the number of circulating endothelial-derived EVs in patients with peripheral arterial occlusive disease [152]. In the same way, Zu el al. [51] showed that lipid-lowering and antihypertensive therapies increased plasma levels of endothelial-derived EVs. Consequently, these EVs reduced the adhesion molecules of monocytes to endothelial cells, such as VCAM-1 and ICAM-1, resulting in improvement of the endothelial function. 

Platelet P2Y12 receptor inhibitors or antagonists, such as clopidogrel and ticagrelor, were suggested to alter the EV counts in plasma. Platelet- and leukocyte-derived EV levels were reported to be lower in patients taking ticagrelor compared to clopidogrel after acute myocardial infarction [153]. The authors suggested that reduction of EVs may explain better clinical outcomes with less thrombotic events in ticagrelor compared to clopidogrel. Chyrchel et al. [154] showed that prasugrel and ticagrelor have higher antiplatelet effect compared with clopidogrel because they decrease plasma levels of platelet-derived EVs. The nitrate supplementation reduced platelet-derived EVs, increasing the response to clopidogrel in CAD patients and it may represent a novel therapeutic strategy to reduce the risk of thrombosis in these patients [155].

## 6. Extracellular Vesicles as Drug Delivery Vehicles in CVD

EVs can incorporate bioactive molecules, act in intercellular communication and have a therapeutic potential, these characteristics have been explored for the use of these vesicles as drug delivery system [13]. EVs may offer high delivery efficiency, intrinsic targeting properties, and low mutagenicity [156]. In addition, the use of EVs as drug delivery vehicle is beneficial as it associates with low immunogenicity, because EVs are biologically produced and have low toxic effects compared with foreign molecules, such as virus-derived vehicles, or cell therapies [157]. Together, these aspects consider the EVs as safe delivery tool. 

For the development of the drug delivery system, the bioactive molecule can be loaded into vesicles during production phase by co-incubation in the cell culture or can be incorporated after the production and isolation of the EVs. Nucleic acids and proteins can be loaded by transfecting the producing cell with the encoding DNA inserted into a vector [158]. 

Based on the ability of EVs to transfer their contents to cells and tissues, circulating EVs involved in cardiovascular protection have been studied, mainly for the delivery of therapeutic miRNAs [158,159]. An *in vitro* study showed that EV-derived cardiac endothelial cells from ischemic myocardium overexpressing hypoxia-inducible factor-1 had higher content of miR-126 and miR-210. These EVs transferred the miRNAs to cardiac progenitor cells and increased the tolerance to hypoxic stress, a protective effect of EVs [160]. 

In apolipoprotein E (apoE)-deficient mice, inhibition of EV-mediated miR-155 transfer from SMC to endothelial cells, using anti-miR-155, reduced the endothelial injury and atherosclerosis, suggesting a promising therapy for atherosclerotic patients [161]. In an animal model (C57BL/6 mice) of myocardial infarction, miRNA-21-loaded EVs were internalized in cardiomyocytes and endothelial cells, restoring the cardiac function [162]. Mesenchymal stem cell-derived EVs were shown to inhibit atherosclerotic plaque formation by delivery of miR-221 to vascular SMC [163]. 

Proteins-derived EVs also have been reported in cardiovascular protection. Vicencio et al [164] demonstrated that EVs loaded with HSP70 had cardioprotective effects in ex vivo, in vivo, and in vitro settings of cardiac ischemia-reperfusion injury. The mechanism involves the stimulation of the toll-like receptor (TLR) 4 by HSP70 and various kinases leading to HSP27 phosphorylation in cardiomyocytes. Leukocyte/platelet-derived EVs were reported to mediate anti-inflammatory effects by downregulation of pro-inflammatory genes [165].

A recent study evaluated the anti-atherosclerotic effect of platelet-derived EVs loaded with MCC950, an NLRP3-inflammasome inhibitor. MCC950-loaded EVs were administrated intravenously and reduced the inflammatory process, the formation of atherosclerotic plaque and inhibited the proliferation of macrophages and T cells in apoE-deficient mice [166]. Decrease of the inflammatory process in atherosclerosis was reported in a study that used molecularly engineered ani-inflammatory M2 macrophage-derived exosomes, and further electroporated with FDA-approved hexyl 5-aminolevulinate hydrochloride (HAL). This study suggested the use of the HAL-engineered M2 macrophage-derived exosomes for atherosclerosis and inflammation-associated diseases therapy [167]. 

The yield of EV isolation is a methodological limitation. A study reported that pH acid (pH4) could be an effective environment to isolate EVs because it increases the levels of EV content, such as RNA and protein (including EV markers), while in alkaline condition (pH11) no EV RNA and proteins have been detected [168]. The yield of EVs can be also affected by storage temperature [169] and solvent (storage buffers, such as Phosphate-buffered saline, Sodium chloride) [157]. The generation of EV mimetics (EVMs) could serve as an important strategy to improve the use of EVs as a novel drug delivery system, using for example to delivery siRNA (siRNA-loaded EVMs) with better yield [156,170,171]. EVMs are vesicles produced artificially from cells or by mixing various lipid compositions similar to EVs. This can be a promising drug delivery vehicle because it maintains the intercellular communication by releasing nucleic acids, proteins, and lipids between cells [172], and can be an important strategy for scale-up production EVs in a short period of time [173].

A previous study suggested that EV uptake in the cells occurs by clathrin-independent endocytosis and micropinocytosis [174]. An important point in the development of efficient EV-based drug delivery therapy is the identification of components on the EV surface that allows their internalization and consequent transfer of their internal cargo to the recipient cells.

To the best of our knowledge, EV delivery approach has not been approved for the treatment of CVD yet. There are more than 50 nanomedicines approved by FDA for some diseases, mainly cancer [175]. These systems use liposomes, polymeric nanoparticles, and inorganic nanoparticles, which have similar to exosome size [176]. EVs as drug delivery system is a technology that offers the opportunity for the development of new pharmacological therapies, but it still needs to be further explored to solve the yield and delivery-associated issues. 

## 7. Conclusions

The great potential of using EVs as a tool to study the pathophysiology of variety of CVD was addressed in this review. The relevance of EVs in intercellular communication and aspects of cellular origin, quantification, and composition of circulating EVs were also explored. Circulating EVs were discussed as potential biomarkers for the diagnosis, prognosis and therapeutic monitoring in CVD, and their risk factors such as metabolic diseases. EVs as biomarkers in CVD seem not so far away to be used in clinic setup. This field is evolving rapidly, and scientists are constantly improving the techniques for isolation, characterization, and analysis of EVs. EVs also have a promising application as a drug delivery system for CVD therapies once technical limitations could be overcome. Future studies on EV composition using more sensitive tools would increase the spectrum of EV clinical applications.

## Figures and Tables

**Figure 1 biomolecules-11-00388-f001:**
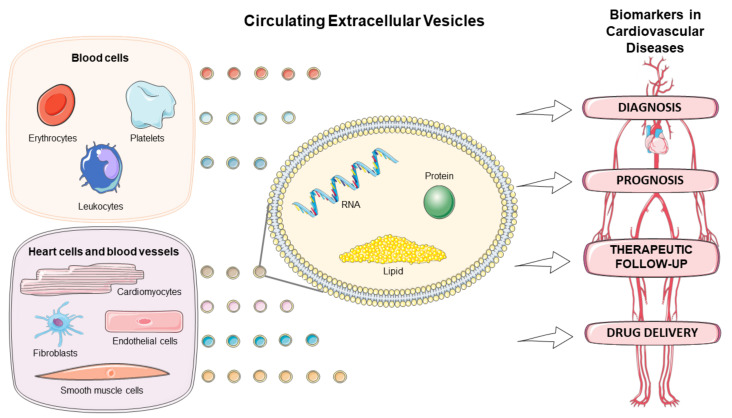
Circulating extracellular vesicles as biomarkers for diagnosis, prognosis, therapeutic follow-up, and drug delivery vehicles in cardiovascular diseases. Figure created using Servier Medical Art images (http://smart.servier.com, accessed on 30 December 2020).

**Table 1 biomolecules-11-00388-t001:** Summary of extracellular vesicles (EV) cargo as biomarkers in cardiovascular disease.

EV Cargo	Source	Disease	EV Isolation	EV Characterization	Quantification Methods	Clinical Outcomes	Ref.
**RNAs**							
lncRNA *Neat1*	Cardiomyocytes	Cardiac ischemia	Ultracentrifugation	Western blot; NTA	qRT-PCR	lncRNA *Neat1* EV modulates the expression of P53 target genes, cell-cycle regulators and promoted cellular survival.	[101]
miR-126 miR-199a	Plasma	CAD	Ultracentrifugation	Flow Cytometry	qRT-PCR	Increased plasma EV miR-126 and miR-199a reduce the risk of major cardiovascular outcomes in CAD patients	[102]
miR-126	Plasma	High-risk CVD	Ultracentrifugation / magnetic beads	TEM; NTA	qRT-PCR	EV miR-126 plasma levels are negatively correlated with NT-proBNP and cTnI. miR-126 as a potential biomarker of CVD	[103]
miR-30 emiR-92a	Plasma	Coronary atherosclerosis	ExoQuick Exosome Precipitation kit (SBI)	−	qRT-PCR	High plasma EV miR-30e and miR-92a, which regulate *ABCA1*, as new biomarkers for clinical diagnosis and treatment of coronary atherosclerosis	[104]
miR-208a	Serum	ACS	ExoQuick Exosome Precipitation kit (SBI)	Western blot	qRT-PCR	Increased serum EV miR-208 is related to early diagnosis and prognosis of ACS	[105]
miR-34a miR-192 miR-194	Serum	HF	ExoQuick Exosome Precipitation kit (SBI)	Western blot	qRT-PCR	Increased serum EV miR-34a, miR-192 and miR-194 are predictive of HF after AMI	[106]
miR-92b-5p	Serum	HF	Exosome isolation kit (RiboBio)	NTA; TEM; Western blot	qRT-PCR	Increased serum EV miR-92b-5p as biomarker for diagnosis of acute HF	[107]
miR-155	Urine	CAD	Ultracentifugation	NTA; TEM; Flow cytometry	qRT-PCR	Increased urinary EV miR-155 as a biomarker of CAD progression	[108]
miR-92a	Endothelial cells	CAD	Ultracentrifugation	Flow cytometry	qRT-PCR	EC-derived EV miR-92a is increased in CAD patients. miR-92a regulates angiogenesis in recipient EC	[109]
miR-92a	Endothelial cells	Atherosclerosis	Ultracentrifugation	TEM; NTA; Western blot	qRT-PCR	EC-derived EV miR-92a as potential therapeutic target in atherosclerosis-related diseases	[110]
miR-128 miR-146 amiR-185 miR-365 miR-503	Macrophages	Atherosclerosis	ExoQuick-TC Exosome Precipitation kit (SBI); Ultracentrifugation	NTA; Western blot	Affymetrix miRNA 3.0 microarray; qRT-PCR	EV-derived miRNAs secreted by atherogenic macrophages may accelerate atherosclerosis	[111]
**Proteins**							
CD31/Annexin 5	Plasma	CAD	PE-conjugated anti-CD31 and FITC-conjugated anti-annexin 5	Flow cytometry	Flow cytometry	Increased plasma CD31/Annexin 5 EVs as an independent predictor of cardiovascular events in CAD patients	[112]
C1Q1A C5 GP1BA PPBP APOD APOC3	Plasma	Myocardial infarction	Ultracentrifugation	Western blot; Cryo-EM	LC-MS/MS	Plasma EV proteins as predictive biomarkers and therapeutic targets in myocardial infarction	[113]
CD144	Plasma	Myocardial injury	Ultracentrifugation	Flow cytometry	Flow cytometry	Increased plasma of CD144-EVs as predictor of cardiovascular complications	[114]
SerpinC1SerpinG1CD14 Cystatin C	Plasma	IHD	Ultracentrifugation	Western blot; TEM; NTA	Bio-plex 200 systems (Bio-Rad)	Plasma EV proteins are associated with stable IHD	[115]
Cystatin C CD14 SerpinG1 SerpinF2	Plasma	HF	OptiPrep™ Density Gradient Medium; Ultracentrifugation	Western blot; TEM	Quantitative Magnetic Bead Assays	Plasma levels of EV CD14, SerpinG1 and SerpinF2 are associated with HF	[116]
Cystatin C pIgR C5a	Serum	ACS	ExoQuick exosome precipitation kit (SBI)	−	Luminex- based multiplex panels	Serum concentrations of EV protein are associated with ACS	[117]
mCRP	Monocytes	CAD	Exo-FLOW^TM^ exosome capture kit	Flow cytometry	Flow cytometry	mCRP in monocyte-derived EVs as biomarker of inflammatory process in CAD patients	[52]
mCRP	Endothelial cells	Myocardial infarction	Ultracentrifugation	Flow cytometry	Western blot; Flow cytometry	EV transport and delivery of pro-inflammatory mCRP in AMI patients	[118]
mCRP	Endothelial cell	PAD	Ultracentrifugation	Flow cytometry; TEM	ELISA; Western blot	EC-derived EV mCRP is increased in patients with PAD, and was suggested as a predictor of vascular disease risk	[119]
ANXA1	Valvular interstitial cells	−	Ultracentrifugation	NTA; TEM; ExoView R100 platform	LC-MS/MS	ANXA1 induces EV aggregation and microcalcification formation and was suggested as a therapeutic target	[120]
CD11b CD16 CD45	Urine	CAD	Ultracentifugation	NTA; TEM; Flow cytometry	Flow cytometry	Increased CD45+ and CD11b+ and decreased CD16+ in urinary EVs are associated with CAD progression	[108]
Nephrin Podocalyxin	Urine	Hypertension	Total Exosome Isolation kit (Invitrogen)	Flow cytometry	Flow cytometry	Urinary levels of EVs enriched in nephrin and podocalyxin are increased in hypertensive patients	[121]
p16	Urine	Hypertension	Total Exosome Isolation kit (Invitrogen)	Flow cytometry	Flow cytometry	Urinary p16 EVs are increased in hypertensive patients	[122]
**Lipids**							
Sphingolipid (ceramides, dihydroceramides, and sphingomyelins)	Plasma	STEMI	Ultracentrifugation	NTA, Flow cytometry; Western blot	LC-MS/MS	EV lipid signature discriminates STEMI patients and may be used as therapeutic strategy	[123]
Phosphatidylserine	Platelet	−	Centrifugation	Flow cytometry; Western blot, TEM	Flow cytometry	EV phosphatidylserine may contribute in thrombin generation and promoting thrombosis	[38]
**Metabolites**							
4-aminohippuric acid Citric acid N-1-methylnicotinamide	Urine	CVD	Ultracentrifugation	TEM; Western blot	SRM-LC-MS/MS	Urinary EV metabolite deregulation as biomarker of CVD	[124]

ABCA1: ATP binding cassette (ABC)A1; ACS: acute coronary syndrome; AMI: acute myocardial infarction; ANXA1: Annexin A1; C5a: complement factor C5a; CAD: coronary artery disease; Cryo-EM: Cryo-electron Microscopy; cTnI: cardiac troponin I; CVD: cardiovascular disease; EC: endothelial cells; FITC: fluorescein isothiocyanate; HF: heart failure; IHD: Ischemic heart disease; LC-MS/MS: liquid chromatography coupled to tandem mass spectrometry; mCRP: pro-inflammatory monomers; NTA: nanoparticle tracking analysis; NT-proBNP: N-terminal propeptide of B-type natriuretic peptide; PAD: peripheral artery disease; PE: Phycoerythrin; pIgR: polygenic immunoglobulin receptor; qRT-PCR: reverse transcription quantitative polymerase chain reaction; SBI: System Biosciences; SRM-LC-MS/MS: Target mass spectrometry in selected reaction monitoring mode, coupled to liquid chromatography; STEMI: ST-segment-elevation myocardial infarction; TEM: transmission electron micrographs.

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
