# Peer review of "Circulating Extracellular Vesicles As Biomarkers and Drug Delivery Vehicles in Cardiovascular Diseases"

_biomolecules, 2021, doi:10.3390/biom11030388_

Round 1
Reviewer 1 Report
Reviewer’s comments and suggestions
The aim of this review was to highlights the role of circulating extracellular vesicles (EVs) in the diagnosis, prognosis, and therapeutic follow-up of CVD, and also for drug delivery. Indeed circulating EVs have attracted in recent years for exploring it as a potential biomarker for diagnostics in several pathological conditions.
Decision: Minor revision
- Line 52-53, need to explore it how?
- Section 2.3 Intercellular communication Not match with the previous section
- Section 3.3, only one study cited? Need to explore it more in this section
- Line 222, In this review, we focused (no need to write like this, please rephrase it)
- Section 4.2 looking at the content of subsection the main section title need to revise
- Line 360, please specify, “circulating EVs derived”
- The author needs to summarize his main focus of study by targeting CVD with EVs, so he should prepare a diagram which describes in every way, need to present one table.
- The author discussed many studies but lack to go inside profoundly to mention the specific point of the other studies
Author Response
Manuscript ID: biomolecules-1080506
We highly value the assessment of the reviewers and addressed all of the concerns in a point-by-point reply below. Our changes in the manuscript have been highlighted in red.
Reviewer 1
The aim of this review was to highlights the role of circulating extracellular vesicles (EVs) in the diagnosis, prognosis, and therapeutic follow-up of CVD, and also for drug delivery. Indeed, circulating EVs have attracted in recent years for exploring it as a potential biomarker for diagnostics in several pathological conditions.
Response: We thank the reviewer 1 for the thoughtful and helpful review of the manuscript.
- Line 52-53, need to explore it how?
Response: We appreciate the reviewer’s comments. As suggested, we explore how the protein markers can be identified and other methods to characterize the EVs. The following specific changes were made in the revised version of the manuscript (Page 2, lines 56-59):
To identify EV’s protein markers, the main methods include Western blotting, ELISA, flow cytometry (FCM) and nano-FCM. In addition, transmission electron microscopy (TEM), dynamic light scattering (DLS) and nanoparticle tracking analysis (NTA) are commonly used [7,11,12].
- Section 2.3 Intercellular communication Not match with the previous section
Response: We agree with the reviewer's comment. We replaced the title of section 2.3 to “EVs interaction between cells from different origins” in the revised manuscript to match the previous section (Page 4, line 159).
- Section 3.3, only one study cited? Need to explore it more in this section
Response: We appreciate reviewer’s suggestion to improve this section. As requested, we added new references on EVs count and heart failure (Page 6, lines 223-227). The following specific changes were made in the manuscript:
Patients with symptoms of chronic HF had increased number of circulating endothelial-derived EVs that was correlated with increased of mortality and recurrent hospitalization risk due to HF [96]. HF patients also had increased serum levels of EVs compared to healthy subjects [97]. A recent review [58] has reported that the number of EVs might be important to differentiate the severity of HF.
- Line 222, In this review, we focused (no need to write like this, please rephrase it)
Response: As suggested, we rephrase it in the revised version of the manuscript (Page 6, line 239-241):
The changes in EV cargo, including RNAs, proteins, and lipids, as potential biomarkers in CVD are reviewed in Table 1.
- Section 4.2 looking at the content of subsection the main section title need to revise
Response: We agree with the reviewer’s suggestion and modified the section title to “Extracellular vesicles carrying proteins” (Page 12, line 325).
- Line 360, please specify, “circulating EVs derived”
Response: We have included the additional information (in red) as requested (Page 14, line 396).
Suades et al. [147] showed a reduction in the number of circulating EVs, specifically mi-croparticles, derived from endothelium, platelets, and inflammatory cells after lipid-lowering therapy with statins.
- The author needs to summarize his main focus of study by targeting CVD with EVs, so he should prepare a diagram which describes in every way, need to present one table.
Response: We appreciate the reviewer’s suggestion. We presented one table with the summary of EV cargo as biomarkers in cardiovascular disease in the revised version of the manuscript (Table 1).
- The author discussed many studies but lack to go inside profoundly to mention the specific point of the other studies
Response: We thank the reviewer for the excellent suggestion to present a table that summarizes the focus of study, which helped us to emphasize the specific points of each studies. Please, see Table 1 in the revised version of the manuscript.
Reviewer 2 Report
The authors review the main findings in EV research in the cardiovascular field, with focus on the application of EVs as biomarkers for both diagnosis and prognosis of CVD and their role as drug delivery vehicles. Although the aim is relevant, I recommend major revision of the manuscript as it recalls one similar recent review in its current form.
Major concerns:
1) the title refers to EVs as biomarkers of CVD but the drug delivery issue is included in the review, it might require reference in the title
2) As mentioned, the authors have reviewed the literature in the field but their work is very similar to the one published by Suet Yen Chong et al, in 2019 (Int. J. Mol. Sci. 2019, 20, 3272; doi:10.3390/ijms20133272). I believe there is room for both works and for an update of the literature in the field, but authors should decide how their work is bringing significant contribution to the field and discuss it. What is the aim of the review and how does it differ from the previous one?
3) Even though referring to the MISEV2018 in the first lines of the manuscript, authors prefer to use the classification of EVs based on the cell origin. I think the introduction should be integrated with the ISEV suggested nomenclature small vs medium/large EVs.
4) nano-flow cytometry is cited as isolation method (page 2 line 44). I would consider it more characterization method than isolation procedure.
5) paragraph 2.2 discusses the role of EVs from heart and blood vessel cells, mentioning their reported role in disease progression or cardioprotection but the issue is only superficially mentioned and should be implemented. How do EV contribute to the cited effects? By releasing miRNA? Proteins? Lipids? Ligand-receptor interaction? Which models have been used to study the cited roles?
6) paragraph 2.3 is entitled intercellular communication. This title should be revised as all EVs and paragraphs deal with intercellular communications, this is the main role of EVs in general
7) pag 4 line 151 authors refer to “specific markers”. Which are they?
8) I find the session dedicated to the use of EV quantification as biomarkers for CVD interesting. Nonetheless authors do not discuss the methodological issue that is crucial in comparing the results of quantification of multiple research groups. How were EV quantified? How were they isolated prior to quantification? The use of EV number instead of antigen profiling is controversial, as well as the standardization of isolation procedures, the pre-analytical treatment of the specimen. There is a lot to say and discuss, please improve.
9) Page 11, lines 425-429: interesting hint about EV isolation yield and influence on EV application in CVD. This point should be implemented.
9) How far we are from the use of EVs as biomarkers in CVD? Please, discuss
Minor comments:
1) “Blood-derived EVs” title is misleading. It should be modified in Blood-cell derived EVs or similar
2) In my opinion subsections in section 3 (EV quantifiction as biomarkers in CVD) are not necessary
Author Response
Manuscript ID: biomolecules-1080506
We highly value the assessment of the reviewers and addressed all of the concerns in a point-by-point reply below. Our changes in the manuscript have been highlighted in red.
Reviewer 2
The authors review the main findings in EV research in the cardiovascular field, with focus on the application of EVs as biomarkers for both diagnosis and prognosis of CVD and their role as drug delivery vehicles. Although the aim is relevant, I recommend major revision of the manuscript as it recalls one similar recent review in its current form.
Major concerns:
Response: We appreciate the reviewer 2 suggestions to improve our manuscript.
- the title refers to EVs as biomarkers of CVD but the drug delivery issue is included in the review, it might require reference in the title
Response: We appreciate the reviewer’s suggestion to include the “drug delivery” in the title. We revised the title to “Circulating extracellular vesicles as biomarkers and drug delivery vehicles in cardiovascular diseases”.
- As mentioned, the authors have reviewed the literature in the field but their work is very similar to the one published by Suet Yen Chong et al, in 2019 (Int. J. Mol. Sci. 2019, 20, 3272; doi:10.3390/ijms20133272). I believe there is room for both works and for an update of the literature in the field, but authors should decide how their work is bringing significant contribution to the field and discuss it. What is the aim of the review and how does it differ from the previous one?
Response: We appreciated the reviewer’s concern. While the topic of our review is somewhat similar to the one published by Suet Yen Chong et al, in 2019, the substantial and rapid increase in the development of EV field around the world and the surge of the numbers of publications each year requires new updates of the field. In addition, our review is different because it addresses the functions attributed to cardiovascular EVs in the diagnostics, prognostics, and therapeutic monitoring of CVD, as well as drug delivery, highlighting recent advances and potential limitations. We also suggested that the EV origin, quantification and content have an important role in CVD.
- Even though referring to the MISEV2018 in the first lines of the manuscript, authors prefer to use the classification of EVs based on the cell origin. I think the introduction should be integrated with the ISEV suggested nomenclature small vs medium/large EVs.
Response: As suggested by the reviewer, we integrated the ISEV nomenclature in the Introduction section (Page 1, lines 35-36). The following specific changes were made in the revised version of the manuscript:
EVs can be classified based on size as small EVs (sEVs), with range < 100 nm or < 200 nm, and medium/large EVs (m/lEVs), with size range > 200 nm [1].
- nano-flow cytometry is cited as isolation method (page 2 line 44). I would consider it more characterization method than isolation procedure.
Response: We fully agree with the reviewer consideration. The nano-flow cytometry allows single-particle measurement of extracellular vesicles, it is not an isolation method and we excluded this method of the introduction section, as suggested.
- paragraph 2.2 discusses the role of EVs from heart and blood vessel cells, mentioning their reported role in disease progression or cardioprotection but the issue is only superficially mentioned and should be implemented. How do EV contribute to the cited effects? By releasing miRNA? Proteins? Lipids? Ligand-receptor interaction? Which models have been used to study the cited roles?
Response: We appreciated the reviewer’s suggestion. We included this information in the section 2.2 (Page 4, lines 124-157). The following specific changes (in red) were made in the revised manuscript:
2.2. Heart cell- and blood vessel-derived EVs
It has been reported that EVs could be released from major cell types in the heart [15,53], such as cardiomyocytes, fibroblasts, endothelial cells, and vascular smooth muscle cells (SMC). Loyer et al. [54] demonstrated, using a murine model of myocardial infarction, that EVs released by cardiomyocytes and endothelial cells following myocardial infarction could be taken up by monocytes and regulate the cardiac inflammatory response by releasing of proinflammatory cytokines.
Endothelial EVs were associated with the progression of atherosclerosis [55], hypertension [51], and CAD [56]. On the other hand, EV can play protective roles. For example, endothelial cell-derived EVs were reported as cardioprotective molecules releasing proteins involved in cellular homeostasis and preservation in the ischemia-reperfusion injury in a chip model of human heart [57]. Several studies were summarized in a review that reported activated endothelial cell-derived EV were also involved in the regulation of cardiac and vascular remodeling in HF [58].
Cardiomyocyte-derived EVs take an important part in the progression of CVD, because they can carry a wide variety of biomolecules, such as proteins and miRNAs, to other cell types and regulate the function and gene expression in these cells [59], especially promoting cardiac repair [60]. Cardiomyocyte-derived EVs secreted from primary cardiomyocytes and human induced pluripotent stem cell-derived cardiomyocytes (hiPSC-CM) can have angiogenic effects after myocardial infarction, through inducing increase expression of miRNAs and proteins, such as growth factors [61,62]; also inducing cardiac fibrosis by release of specific miRNAs via myocyte-fibroblast cross-talk [63]. EVs released from cardiomyocytes derived from human-induced pluripotent stem cells were also used in treatment of heart injury, including myocardial infarction, contributing to cardiac regeneration, through cardiac-specific miRNAs activity [64,65].
Cardiac fibroblast-derived EVs stimulated cell cardiac migration [66], SMC proliferation, and vascular remodeling [67] by release of miRNAs. Vascular SMC-derived EVs are enriched with RNAs, proteins and lipids associated with vascular remodeling, calcification and coagulation [68–72], familial hypercholesterolemia and CAD [73]. Most studies have evaluated changes in SMC caused by EVs derived from other cells. For example, EVs derived from bone marrow mesenchymal stem cells (BMSC) have been shown to induce calcification in vascular SMC by modifying miRNA profiles [74], and EV derived from platelets could modulate inflammatory response in vascular SMC by presenting chemokine CXCL4 and membrane-bound effectors [75].
- paragraph 2.3 is entitled intercellular communication. This title should be revised as all EVs and paragraphs deal with intercellular communications, this is the main role of EVs in general
Response: We appreciate the reviewer's recommendation. We changed the title of paragraph 2.3 to “EVs interaction between cells from different origins” in the revised manuscript (Page 4, line 159)
- pag 4 line 151 authors refer to “specific markers”. Which are they?
Response: The specific markers are used in the study cited for the discrimination of EV origin cells (platelets, monocytes, granulocytes, T cells, and NK cells). The authors used CD41 as marker for platelet origin, CD45-PB and CD14-PE as monocyte markers, CD16/56-PC5 as granulocyte and NK cell marker, and CD3-ECD as T cell marker. To clarify this point, we added more information (red) in the revised version of the manuscript (Page 5, lines 166-168):
Weiss et al. [76] reported differential interaction of platelet-derived EVs with monocytes and other leukocytes, which were identified by specific markers using flow cytometry. The authors used CD41 as marker for platelet origin, CD45-PB and CD14-PE as monocyte markers, CD16/56-PC5 as granulocyte and NK cell marker, and CD3-ECD as T cell marker.
- I find the session dedicated to the use of EV quantification as biomarkers for CVD Nonetheless authors do not discuss the methodological issue that is crucial in comparing the results of quantification of multiple research groups. How were EV quantified? How were they isolated prior to quantification? The use of EV number instead of antigen profiling is controversial, as well as the standardization of isolation procedures, the pre-analytical treatment of the specimen. There is a lot to say and discuss, please improve.
Response: We appreciate reviewer comments and considered these important observations. As suggested, we included the discussion about the methodological issue in the section 3 (Page 5, lines 186-189). Please, see below in red the specific changes in the revised version of the manuscript:
Several studies have shown an association of circulating EV counts with CVD, suggesting a potential application of EV quantification as a biomarker for diagnostic and therapeutic monitoring [25,82,83]. Although using to count of EV from particular cell type as biomarker seems promising, the major limitation of this approach is the lack of standardization of measurements, resulting in difficulty to compare studies from multiple research groups [25].
- Page 11, lines 425-429: interesting hint about EV isolation yield and influence on EV application in CVD. This point should be implemented.
Response: We appreciate reviewer’s pointing out this important aspect. As suggested, we implemented this point in the revised manuscript (Page 15, lines 461-471). The following specific changes (in red) were made in the revised version of the manuscript:
The yield of EV isolation is a limitation, a study reported that pH acid could be an effective environment to isolate EVs because it increases the levels of EV content, such as RNA and protein (including EV markers), while in alkaline condition in EV RNA and proteins have been detected [164]. The yield of EV can be also affected by storage temperature [165] and solvent (storage buffer) [153]. The generation of EV mimetics (EVMs) is an important strategy to improve the use of EVs as a novel drug delivery system, using for example to delivery siRNA (siRNA-loaded EV-mimetics) with better yield [152,166,167]. EVMs are vesicles produced artificially from cells or in combination with lipid materials. This can be promising drug delivery vehicle because it maintains the intercellular communication by releasing nucleic acids, proteins and lipids [168], and can be an important strategy for scale-up production EVs in a short period of time [169].
- How far we are from the use of EVs as biomarkers in CVD? Please, discuss
Response: We appreciate the reviewer’s comments. The use of EVs as biomarkers in CVD seems not so far away. This scientific field is rapidly evolving, and scientists are constantly improving the techniques for isolation, characterization, and analysis of EVs. EVs also have a promising application as a drug delivery system for cardiovascular diseases therapies once some technological limitations could be overcome. More studies in composition and functions using more sensitive tools can increase the spectrum of EVs application. We added this comment in the conclusion section (Page 16, lines 491-496) as followed (in red):
EVs as biomarkers in CVD seem not so far away to be used in clinic. This field is evolving rapidly, and scientists are constantly improving the techniques for isolation, characterization, and analysis of EVs. EVs also have a promising application as a drug delivery system for cardiovascular disease therapies once some technological limitations could be overcome. More studies on EV composition using more sensitive tools would increase the spectrum of EV clinical applications.
Minor comments:
- “Blood-derived EVs” title is misleading. It should be modified in Blood-cell derived EVs or similar
Response: We agree with the reviewer. As suggested, we modified “Blood-derived EVs” to “Blood-cell derived EVs” in section 2.1 (Page 3, line 97) in the revised version of the manuscript.
- In my opinion subsections in section 3 (EV quantification as biomarkers in CVD) are not necessary
Response: We thank the reviewer for this suggestion. The subsections in section 3 have been removed in the revised version of our manuscript.
Round 2
Reviewer 2 Report
My concerns and suggestions have been addressed and authors have implemented their work that is now suitable for publication.
Author Response
Thank you